# Palm up: Playing in the Latent Manifold for Unsupervised Pretraining

**Hao Liu**[*]
UC Berkeley

**Tom Zahavy**
DeepMind

**Volodymyr Mnih**
DeepMind

**Satinder Singh**
DeepMind

## Abstract

Large and diverse datasets have been the cornerstones of many impressive advancements in artificial intelligence. Intelligent creatures, however, learn by interacting with the environment, which changes the input sensory signals and the state of the environment. In this work, we aim to bring the best of both worlds and propose an algorithm that exhibits an exploratory behavior whilst it utilizes large diverse datasets. Our key idea is to leverage deep generative models that are pretrained on static datasets and introduce a dynamic model in the latent space. The transition dynamics simply mixes an action and a random sampled latent. It then applies an exponential moving average for temporal persistency, the resulting latent is decoded to image using pretrained generator. We then employ an unsupervised reinforcement learning algorithm to explore in this environment and perform unsupervised representation learning on the collected data. We further leverage the temporal information of this data to pair data points as a natural supervision for representation learning. Our experiments suggest that the learned representations can be successfully transferred to downstream tasks in both vision and reinforcement learning domains.

## 1 Introduction

Large and diverse datasets have been the cornerstones of many impressive successes at the frontier of artificial intelligence, such as protein folding [64, 28], image recognition [55, 13], and understanding natural language [6, 12]. Training machine learning models on diverse datasets that cover a breadth of human-written text and natural images often dramatically improves performance and enables impressive generalization capabilities [16, 6, 55, 12]. These models are learned from a fixed set of images, videos, or languages, guided by supervision that comes from ground-truth labels [13, 55], self-supervised contrastive learning [10], or masked token prediction [6, 12] among other approaches.

In contrast, intelligent creatures learn to perform interactions that actively change the input sensory signals and the state of the environment towards a desired configuration. For example, in psychology, it has been shown that interaction with the environment is vital for developing flexible and generalize intelligence [23, 74, 17]. During the first few months of interactions, infants develop meaningful understandings about objects [67] and prefer to look at exemplars from a novel class (e.g., dogs) after observing exemplars from a different class (e.g., cats) [54]. We hypothesis that by situating the agent in a high semantic complexity environment, the agent can develop interesting cognition abilities.

Given the effectiveness of learning from large and diverse datasets and the significance of interactive learning behaviors in intelligent creatures, it is therefore extremely important to connect them to build better learning algorithms.

To bridge the gap, perhaps one straightforward solution is learning in the diverse real world [1, 53], however, teaching a robot to interact in the physical world is both time consuming and resource

---

[*]This work was partially done at DeepMind. Correspondence to `hao.liu@cs.berkeley.edu`

36th Conference on Neural Information Processing Systems (NeurIPS 2022).

intensive, and it is also difficult to scale up. An alternative is learning in simulated environments which have got surged interest recently. There simulators, such as Habitat [61, 70], RLBench [27], House3D [77], and AI2THOR [36] enable the agent to interact with its environment. However, the visual complexity of these simulated environments is far from matching the intricate real world. The key limitation is that making hand-designed simulation that is close enough to what a camera in the real-world would capture is both challenging and tedious.

To remedy the issue, we propose a conceptually simple yet effective method that leverages existing diverse datasets, builds an environment with high semantic complexity from them, and then performs interactive learning in this environment. We do so by leveraging deep generative models that are trained in static datasets and introduce transition dynamics in the latent space of the generative model. Specifically, at each time step, the transition dynamics simply mix action and a random sampled latent. It then applies an exponential moving average for temporal persistency, imitating the prevalent temporal persistency in the real world. Finally, the resulting latent is decoded to an image using a trained generator. The generator is a conditional generative model which is conditioned on a prompt $e.g.$ a class label sampled at the beginning of an episode for achieving further temporal persistency. For the generative model, we use conditional StyleGAN [32] in this work, chosen for its simplicity, although our method is not restricted to it, and can be also applied to other generative models such as language conditioned model Dalle [56].

We employ unsupervised reinforcement learning (RL) [41] to explore this environment motivated by imitating how intelligent creatures acquire perception and action skills by curiosity [23, 35, 60]. Specifically, we use the nonparametric entropy maximization method named APT [44] which encourages the agent to actively explore the environment to seek novel and unseen observations. Similar to other pixel-based unsupervised RL methods, APT learns an abstract representation by using off-the-shelf data augmentation and contrastive learning techniques from vision [37, 40, 39]. While effective, designing these techniques requires domain knowledge. We show that by simply leveraging the temporal nature, representation can be effectively learned. We do so by maximizing the similarity between representations of current and next observations based on siamese network [5] without needing to use domain knowledge or data augmentation. Our method is named as **pl**aying in the **l**atent **m**anifold for unsupervised pretraining (PALM).

We conduct experiments in CIFAR classification and out-of-distribution detection by transferring our unsupervised exploratory pretrained representations in StyleGAN-based environments. Our experiments show that the learned representations achieve competitive results with state-of-the-art methods in image recognition and out-of-distribution detection despite being only trained in synthesized data without data augmentation. We also train StyleGAN in observation data collected from Atari and apply our method to it. We found that the learned representation helped in maximizing many Atari game rewards. Our major contributions are summarized below:

- We present a surprisingly simple yet effective approach to leverage generative models as an interactive environment for unsupervised RL. By doing so, we connect vision datasets with RL, and enable learning representation by actively interacting with the environment.

- We demonstrate that exploration techniques used in unsupervised RL incentivize RL agent to learn representations from a synthetic environment without data augmentations.

- We show that PALM matches SOTA self-supervised representation learning methods on CIFAR and out-of-distribution benchmarks.

- We show that PALM outperforms strong model-free and model-based training from scratch RL. It also achieves competitive scores as SOTA exploratory pre-training RL and offline-data pretraining RL methods.

## 2  Related work

**Exploratory pretraining in RL**   Having an unsupervised pretraining stage before finetuning on the target task has been explored in reinforcement learning to improve downstream task performance. One common approach has been to allow the agent a period of fully-unsupervised interaction with the environment, during which the agent is trained to maximize a surrogate exploration-based task such as the diversity of the states it encounters [44, 43, 78]. Others have proposed to use self-supervised objectives to generate intrinsic rewards encouraging agents to visit new states, such as the loss of an

inverse dynamics model [52, 7]. SGI [63] combines forward predictive representation learning [62] with inverse dynamics model [52] and demonstrate the power of representation pretraining for downstream RL tasks. Massive-scale unsupervised pretraining has shown strong results [8]. Laskin et al. [41] conducted a comparison of different unsupervised pretraining reinforcement learning algorithms. Finally, Chaplot et al. [9], Weihs et al. [75] studied training RL agent in game simulators and transferring its representation to various vision tasks. Their environments are equipped with carefully chosen domain-specific reward function to guide the learning of RL agent, and the architectures of their RL agents are fairly complicated.

Our work differs in that we do not rely on hand-crafted simulators and renderers which require a huge amount of domain knowledge and effort to build, instead we leverage generative models as renders. Unlike many prior work in unsupervised pretraining RL, our work does not focus on improving transfer performance to downstream RL tasks although it can be used for this purpose.

**Training with synthetic data**  Using deep generative models as a source of synthetic data for representation learning has been studied in prior work [59, 26, 33]. These generative models are fit to real image datasets and produce realistic-looking images as samples. Baradad et al. [2] studied using data sampled from random initialized generative models to train contrastive representations. Gowal et al. [20] studied combining data sampled from pretrained generative models with real data for adversarial training and demonstrated improved results in robustness. The use of synthesized data has been explored in reinforcement learning under the heading of domain randomization [72], where 3D synthetic data is rendered under a variety of lighting conditions to transfer to real environments where the lighting may be unknown. Our approach does away with the hand crafted simulation engine entirely by making the training data diverse through unsupervised exploration.

Different from them, our work focus on leveraging a generative model as an interactive environment and learn representation without using data augmentation.

**Temporal persistent representation**  Using temporal persistent information for representation learning has been proposed in the past with similar motivations as ours. It has been used in learning representation from videos [3, 76, 48, 19, 14, 51] by minimizing different metrics of representation difference over a temporal segment. Learning persistent representation has been explored in reinforcement learning, and has been demonstrated to improve data efficiency [50, 65, 62, 79] and improve downstream task performance [68, 63].

In relation to these prior efforts, our work studies visual representation learning from interacted experiences based on real-world data.

## 3   Preliminary

**Unsupervised reinforcement learning**  Reinforcement learning considers the problem of finding an optimal policy for an agent that interacts with an uncertain environment and collects reward per action [69]. The agent maximizes its cumulative reward by interacting with its environment.

Formally, this problem can be viewed as a Markov decision process (MDP) defined by $(\mathcal{S}, \mathcal{A}, \mathcal{T}, \rho_0, r, \gamma)$ where $\mathcal{S} \subseteq \mathbb{R}^{n_s}$ is a set of $n_s$-dimensional states, $\mathcal{A} \subseteq \mathbb{R}^{n_a}$ is a set of $n_a$-dimensional actions, $\mathcal{T} : \mathcal{S} \times \mathcal{A} \times \mathcal{S} \rightarrow [0, 1]$ is the state transition probability distribution. $\rho_0 : \mathcal{S} \rightarrow [0, 1]$ is the distribution over initial states, $r : \mathcal{S} \times \mathcal{A} \rightarrow \mathbb{R}$ is the reward function, and $\gamma \in [0, 1)$ is the discount factor. At environment states $s \in \mathcal{S}$, the agent take actions $a \in \mathcal{A}$, in the (unknown) environment dynamics defined by the transition probability $T(s'|s, a)$, and the reward function yields a reward immediately following the action $a_t$ performed in state $s_t$. In value-based reinforcement learning, the agent learns an estimate of the expected discounted return, a.k.a, state-action value function $Q^\pi(s_t, a_t) = \mathbb{E}_{s_{t+1}, a_{t+1}, \ldots} \left[ \sum_{l=0}^{\infty} \gamma^l r(s_{t+l}, a_{t+l}) \right]$. A new policy can be derived from value function by acting $\epsilon$-greedily with respect to the action values (discrete) or by using policy gradient to maximize the value function (continuous).

In unsupervised reinforcement learning, the reward function is defined as some form of intrinsic reward that is agnostic to standard task-specific reward function $r := r_{\text{intrinsic}}$. The intrinsic reward function is usually constructed for a better exploration and is computed using states and actions collected by the agent.

**Generative adversarial model** Generative Adversarial Networks (GANs) [18] consider the problem of generating photo realistic images. The StyleGAN [30, 32, 31] architecture is one of the state-of-the-art in high-resolution image generation for a multitude of different natural image categories such as faces, buildings, and animals.

To generate high-quality and high-resolution images, StyleGAN makes use of a specialized generator architecture which consists of a mapping network and synthesis network. The mapping network converts a latent vector $z \in \mathcal{Z}$ with $\mathcal{Z} \in \mathbb{R}^n$ into an intermediate latent space $w \in \mathcal{W}$ with $\mathcal{W} \in \mathbb{R}^n$. The mapping network is implemented using a multilayer perceptron that typically consists of 8 layers. The resulting vector $w$ in that intermediate latent space is then transformed using learned affine transformations and used as an input to a synthesis network.

The synthesis network consists of multiple blocks that each takes three inputs. First, they take a feature map that contains the current content information of the image that is to be generated. Second, each block takes a transformed representation of the vector $w$ as an input to its style parts, followed by a normalization of the feature map.

## 4 Method

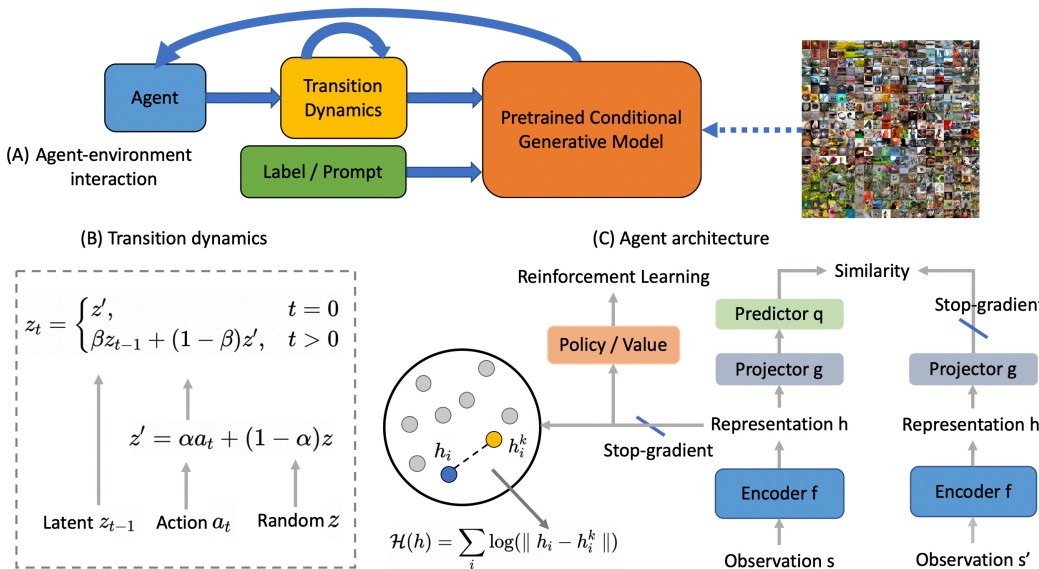

Figure 1: Overview of the proposed method. **(A)**. A conditional generative model is pretrained in a static dataset which is conditioned a prompt (*e.g.* language or class label) sampled at the beginning of an episode, and a transition dynamics is defined in the latent space of the generator, the agent maximizes the nonparameteric entropy of experience in learned representation space. **(B)**. The transition dynamics consists of mixing a randomly sampled latent with action (of the same dimension) followed by exponential moving average for temporal persistency, the resulting latent is decoded to image using pretrained generator. **(C)**. The representation is learned without using data augmentation by maximizing the representation similarity between two consecutive synthesized observations based on Siamese network. The agent is updated using unsupervised reinforcement learning with representation detached.

Our objective is to leverage pretrained deep generative models $G : \mathcal{Z} \times \mathcal{C} \to \mathcal{S}$ where $\mathcal{Z}$ denote latents, $\mathcal{C}$ denote prompt or labels and $\mathcal{S}$ denote observations to build an interactive environment and train an unsupervised reinforcement learning agent in such environment for representation pretraining.

### 4.1 Latent environment dynamics

The transition dynamics are designed in the latent space of StyleGAN. At the beginning of an episode, a class label or prompt $c$ is randomly sampled, at each time step, $c$ and a latent $z_t$ that depends on the action and previous latent $z_t = T(a, z_{t-1})$ are transformed by the synthesis network of StyleGAN into an image which serves as an observation $s_t = G(z_t, c)$. We note that while the environment conditions on the label, the ground-truth information is not directly used by PALM.

The transition dynamics model consists of two steps. In the first step, it combines a randomly sampled latent $z$ with the agent's action to obtain a new latent $z' = T_{\text{com}}(z, a_t)$. Although more advanced combination methods can be used, we use simple weighted summation to combine latent and action $T_{\text{com}}(z, a_t) = \alpha a_t + (1 - \alpha)z$, where $a$ is an action generated by the policy and $\alpha \in [0, 1]$ is a hyperparameter controls the relative importance of action and latent.

The second step is a transformation $T_{\text{ema}}(z')$ which is designed to ensure temporal persistency of the environment. We use a simple exponential moving average (EMA) to implement $T_{\text{ema}}$. Specifically, this module generate new latent by following

$$z_t = \begin{cases} z', & t = 0 \\ \beta z_{t-1} + (1 - \beta)z', & t > 0 \end{cases}$$

where $z'$ is the combined latent and action, $z_t$ is the exponential moving average at timestep t, and $\beta \in [0, 1]$ is a hyperparameter that controls how fast the environment changes in respond to agent. We used $\beta = 0.95$ in most of our experiments per our initial experiments.

The high $\beta$ is motivated by observations in unsupervised RL and linguistics. It has been shown that a highly random environment hinders the learning of unsupervised RL agents which is also known as the "Noisy-TV" problem [7]. It also has an interesting connection to the uniform information density hypothesis in linguistics which states that—subject to the constraints of the grammar—humans prefer sentences that distribute information equally across the linguistic signal, *i.e.*, evenly distributed surprise [42].

In addition to the latent $z_t$, we also condition the generator $G(z, c)$ on a random sample class label $c$ sampled at the beginning of an episode, resulting in further persistent consistency. Finally, the environment dynamics can be written as $T := \mathcal{G}(T_{\text{ema}}(T_{\text{com}}(z, a)), c)$.

## 4.2 Temporal persistent representation

Our goal is to pretrain a representation $f : \mathcal{S} \rightarrow \mathcal{H}$ which is a mapping that maps observations $s$ to lower dimensional representations $h$, using the data (experience replay) collected by the agent. Data augmentations are key ingredients of unsupervised representation learning methods [see e.g. 10, 11]. However, designing augmentations requires domain knowledge.

Our representation learning is instead based on temporal information. Our key idea is that since each trajectory is generated by interacting with a deep generative model conditions on the same label and temporal persistent latent, it provides natural augmentations of visual content under various changing factors, such as occlusion and illumination. Some rollout trajectories during training are presented in Figure 2. In addition, the exploratory learning behavior of the agent increases the diversity of the trajectory.

Specifically, the idea is to train the encoder $f(s)$ to produce embeddings that are persistent in time in the same trajectory, *i.e.*, two presentations $h_t = f(s_t)$ and $h_{t+1} = f(s_{t+1})$ should be similar. To do so, we use the Siamese network [5] and its modern training techniques from SimSiam [11]. Specially, we maximize the similarity between $q(g(f(s)))$ and `stop_gradient`$(g(f(s')))$ where q and g are multilayer perceptrons. More details can be found in appendix.

## 4.3 Unsupervised exploration

The objective of unsupervised exploration is to maximize the diversity of the data seen by the agent, imitating how intelligent creatures develop recognition.

An observation $s_t$ is first encoded using the encoder $f$ to get representation $h_t = f(s_t)$. The policy takes representation $h_t$ as input and generates action $a_t$ of the same dimension. The policy distribution is a tanh Gaussian following prior work in unsupervised reinforcement learning [78, 41].

We resort to maximizing the nonparametric entropy of the representation space which has been widely used in unsupervised reinforcement learning [66, 4, 44, 78]. By explicitly encouraging the agents to visit diverse states, it effectively encourages the agents to explore environments, and thus learn more diverse behaviors. The estimator computes the distance between each particle $h_i = f(s_i)$ and its $k$-th

nearest neighbor $h_i^\star$.

$$H(h) = \sum p(h) \log p(h) \propto \sum_{i=1}^{n} \log \|h_i - h_i^\star\|_2,$$

where $\| \cdot \|_2$ is $\ell_2$ norm. We associate each transition from experience replay $\{(s, a, s')\}$ with an intrinsic reward given by

$$r_{\text{intrinsic}}(s, a, s') = \log \|h - h_i^\star\|_2.$$

Putting everything together, the diagram of our method is shown in Figure 1.

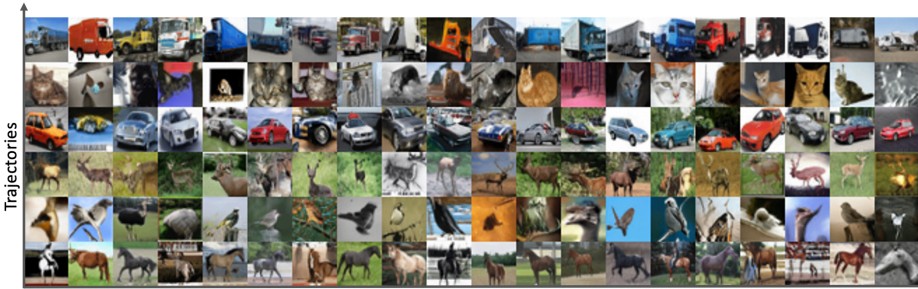

Figure 2: Multiple rollouts collected by interacting with environment in PALM. Each row represents a randomly sampled segment of a trajectory. From top to bottom, each trajectory is conditioned on a different class label $c$.

## 5 Results

It is important to stress that this work focuses on studying how combined exploratory pretraining and static datasets contribute to representation learning in artificial agents and not on developing a new, state-of-the-art, methodology for representation learning. Nevertheless, to better situate our results in the context of existing work, we provide strong baselines in our experiments.

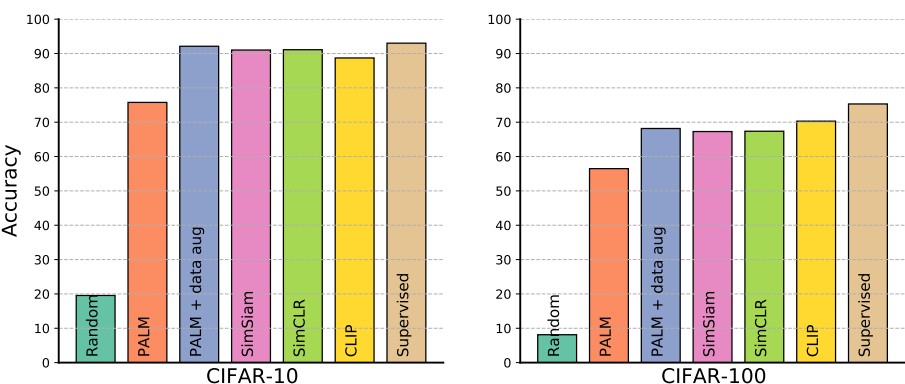

Figure 3: Linear classification results from different methods in CIFAR datasets. Model is ResNet-18 except for CLIP [55] which is based on ResNet-50. Supervised ResNet [22] is trained from scratch. SimCLR [10] and SimSiam [11] are pretrained on CIFAR dataset. CLIP [55] is pretrained on paired image-text datasets. We use reported scores of each baseline for comparison. All results of our method are the average of three runs with different seeds. PALM (our method) actively collect synthesized data from deep generative models based environment. The representation is learned without using data augmentation from explored experience but still achieves competitive results. Our PALM, which is based on SimSiam [11], is compatible with data augmentation, by adding the same data augmentation used in SimSiam, PALM + data aug matches or outperforms state-of-the-art static representation learning algorithms.

**Image classification** We test PALM on image classification tasks using a ResNet-18 as encoder $f$. The RL agent interacts with the environment for 50M steps, and trains representation using the procedure described in the method section. The hyperparameters of the representation learning follow the default hyperparameters of SimSiam [11] and more details can be found in the supplemental material.

We compare our method with a series of strong methods in static datasets, including state-of-the-art contrastive representation learning methods SimCLR [10] and SimSiam, supervised trained model, and CLIP [55] which is pretrained on a large-scale paired image-text datasets.

Figure 3 shows the linear classification performance of each method, PALM is surprisingly effective given the lack of data augmentation and real data. Note that the baselines do not have an option not to use data augmentation.

Additionally, adding the same data augmentation techniques used in SimSiam to our method further improves the results, matching or outperforming current SOTA static representation learning algorithms. Compared with the SOTA method SimSiam, PALM + data augmentation actively explore an environment instead of using a static dataset, and the representation learning and data augmentation are applied to synthesized temporal pair instead of two views of the same image.

Table 1: AUROC (%) of classifiers trained on CIFAR-10. Models are ResNet-18. The reported results of PALM are averaged over five trials. To have a rigorous comparison, the results of baselines are from their papers. A → B denotes in-distribution dataset A and out-of-distribution dataset B. Subscripts denote standard deviation, and bold denote the best results.

| | CIFAR10 → | | | | |
|---|---|---|---|---|---|
| Train method | SVHN | LSUN | CIFAR100 | Interp. | CelebA |
| Standard [24] | $88.6_{\pm 0.9}$ | $90.7_{\pm 0.5}$ | $85.8_{\pm 0.3}$ | $75.4_{\pm 0.7}$ | $64.1_{\pm 0.5}$ |
| SupCLR [34] | $97.3_{\pm 0.1}$ | $92.8_{\pm 0.5}$ | $88.6_{\pm 0.2}$ | $75.7_{\pm 0.1}$ | $73.2_{\pm 0.3}$ |
| CSI [71] | $96.5_{\pm 0.2}$ | $\mathbf{96.3_{\pm 0.5}}$ | $90.5_{\pm 0.1}$ | $78.5_{\pm 0.2}$ | $75.1_{\pm 0.3}$ |
| PALM (ours) | $\mathbf{97.9_{\pm 0.2}}$ | $96.1_{\pm 0.3}$ | $\mathbf{91.5_{\pm 0.2}}$ | $\mathbf{79.9_{\pm 0.2}}$ | $\mathbf{76.8_{\pm 0.4}}$ |

**Out-of-distribution detection** While some prior work demonstrated that self-supervised learning approaches significantly improve OOD detection performance [25, 15], their self-supervised pre-train heavily relies on domain-specific data augmentations, we want to study how PALM performs on OOD benchmarks.

We compare PALM with standard supervised model [22, 24] and contrastive representation learning based methods including SupCLR [34] and CSI [71]. CSI demonstrates state-of-the-art results by combining contrastive representation learning and supervised learning. Table 1 shows the comparison on multiple datasets. PALM outperforms CSI in 4 out of 5 out-of-the-distribution datasets, despite using only synthesized data and no data augmentation. We attribute the effectiveness of PALM on OOD detection to active exploration stimulates the model to learn the invariance of data. In doing so, the representation is optimized for distinguishing different observations.

**Data efficient representation in RL** In this experiment, we are interested in testing PALM pre-trained model for downstream reinforcement learning tasks. To do so, we focus our experimentation on the Atari 100k benchmark introduced by Kaiser et al. [29], in which agents are allowed only 100k steps of interaction with their environment.

There has been extensive usage of Atari for representation learning and exploratory pretraining [8, 44, 63, 21]. Finetuning after exploratory pretraining is highly effective [e.g., 63, 8], providing strong baselines to compare to.

Unlike experiments in CIFAR datasets, here we train an unconditional StyleGAN model in a dataset collected by running a double DQN [47, 73] agent. The dataset consists of 50M frames for each game. We use 6M steps for exploratory pretraining in the synthesized environment and we utilize DrQ [37] for online finetuning. Table 2 shows the comparison of PALM with baselines. Comparing

Table 2: HNS on Atari100k for PALM and baselines. >H denotes super-human performance and >0 denotes greater than random performance. Mdn and mean denote the median and mean scores. Bold denote the best results in each category.

| Method | Mdn | Mn | >H | >0 | Data |
|---|---|---|---|---|---|
| *No Pretraining (Finetuning Only)* | | | | | |
| SimPLe | 0.144 | 0.443 | 2 | **26** | 0 |
| DrQ | 0.268 | 0.357 | 2 | 24 | 0 |
| *Exploratory Pretraining + Finetuning* | | | | | |
| APT | **0.475** | 0.666 | 7 | **26** | 250M |
| ATC | 0.237 | 0.462 | 3 | **26** | 3M |
| SGI | 0.456 | **0.838** | 6 | **26** | 6M |
| *Offline-data Pretraining + Finetuning* | | | | | |
| ATC | 0.219 | 0.587 | 4 | **26** | 3M |
| SGI | **0.753** | **1.598** | 9 | **26** | 6M |
| *Synthesized Exploratory Pretraining + Finetuning* | | | | | |
| PALM (ours) | **0.298** | **0.411** | 2 | **26** | 50M + 6M |

with exploratory pretraining baselines (APT [44], ATC [68], and SGI [63]) that are pretrained in Atari games directly, PALM achieves surprisingly good results considering that it is trained in a simple synthesized environment. PALM achieves higher medium and mean score than DrQ in Atari 100k despite the online finetuning of PALM is the same as DrQ, demonstrating the benefit of pretrained representation. Compared with offline-data pretraining baselines (ATC and SGI), there is still a gap between PALM and the state-of-the-arts, which is probably due to PALM's environment being trained on observations only while ATC and SGI are trained on both observations and actions in a model-based style. Worth mentioning that while PALM utilizes 50M offline action-free experience for pretraining, the amount is less than the 250M online action labeled experience used in APT. We would like to emphasize that DrQ is based on data augmentation, but PALM can outperform DrQ without data augmentation given the same amount of online experience.

## 6 Analysis

**Persistent environment and active interaction are important.** Next, we perform an ablation study, to study the importance of a persistent environment. We aim to answer the following question. How important it is to have action (controlled by $\alpha$) and state (controlled by $\beta$) in the transition dynamics? When $\beta = 0$, there is effectively no state, $i.e.$, its a bandit. As shown in Figure 4, when decreasing the importance of states which is denoted as PALM ($\alpha = 0.5$ $\beta = 0.5$), the performance decreases. The extreme case $\beta = 0$ shows a significant drop in accuracy. These results indicate that it is important to have temporal persistent transition dynamics. When $\alpha = 0$, action is ignored in

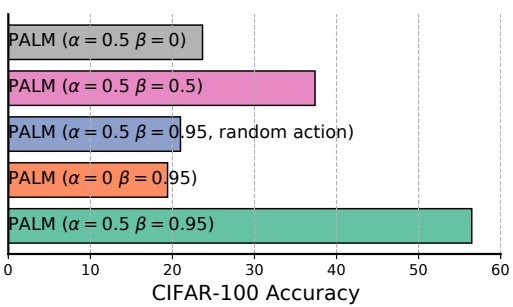

Figure 4: Ablative analysis of PALM on CIFAR-100 dataset. All results are the average of three runs with different seeds.

transition dynamics, this ablative baseline performs significantly worse than its default $\alpha = 0.5$. When replacing the agent's action with random noise which is denoted as PALM + random action, we also see a significant performance drop. These results show having active interaction is important to the results of PALM. We present the results of more combinations of $\alpha$ and $\beta$ in Figure 5, the results show that as $\alpha$ decreases from 0.95 to 0, because of the decreasing importance of action, the accuracy continues to drop. $\alpha = 0.95$ in general performs better than $\alpha = 1.0$, indicates mixing a small amount of random noise with action helps exploration and improves representation learning. When $\alpha$ or $\beta$ is zero, the results are significantly worse. We also see that $\beta \in [0.5, 0.75]$ performs significantly better than other values, showing that persistence of dynamics is crucial.

**Exploratory Learning as Curriculum Learning.** As we have seen, representations learned from exploratory learning in a synthesized environment can be surprisingly powerful. We are interested in having a perhaps preliminary study of the reasons.

To do so, we compare PALM with its passive version (passive PALM), static version (static PALM), and methods proposed in Jahanian et al. [26]. In the random variant, the positives pairs are sampled from $G(z, c = c')$ (i.e. fix the class, take two random draws from z). We denote this baseline as Random Jahanian et al. In Gaussian Jahanian et al, the positive pairs are given by z and z+w, where w is Gaussian.

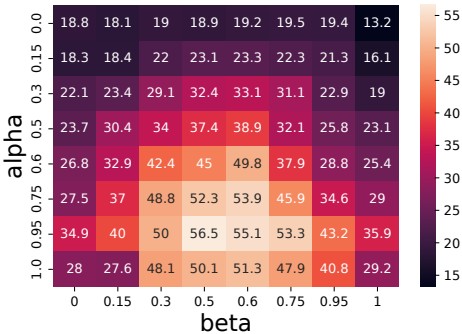

Figure 5: Ablative analysis of PALM on CIFAR-100 dataset. All results are the average of three runs with different seeds.

Static PALM is not trained in the interactive environment, instead, it is trained with real images. Specifically, we randomly sample two images of the same label from a dataset and use them as a temporal pair to train the representation learning of PALM. We note that static PALM is related to SupCon [34], but it differs in that static PALM does not use any data augmentation while SupCon leverages a

combination of data augmentations. SupCon also uses a large number of negative examples from different categories.

Passive PALM is trained offline on the experience of PALM. We first store all the online experience (50M in total) of PALM, then we use the saved experience to train passive PALM without having to interact with the environment. We consider two options of passive PALM. The first one is random passive PALM which randomly samples consecutive observations from the stored offline experience. The second one is ordered passive PALM which is trained on the same order of observations as in online experience.

With the number of data increasing, PALM keeps improving with the amount of training data and significantly outperforms static PALM even though static PALM is trained on real datasets. We believe the reason is that the diverse synthesized data collected by PALM can greatly help to learn representation. In the low data regime, carefully designed non-active learning based methods [26] outperforms PALM, possibly due to RL training being sample inefficient. With the increasing number of synthesized samples, PALM continues to improve and significantly outperform all baselines. The results also show that PALM scales better than non active learning based methods. PALM nearly always outperforms random passive PALM, demonstrating the effectiveness of exploratory learning. Interestingly but not surprisingly, ordered passive PALM performs nearly the same as PALM, indicating the exploratory learning behavior of PALM might play an important role in curriculum learning. The agent is optimized to search for novel samples in terms of their representation, by doing so, the searched samples should become more difficult to distinguish. We believe a comprehensive understanding of the effectiveness of PALM is out of the scope of this work, we tend to leave it as future work.

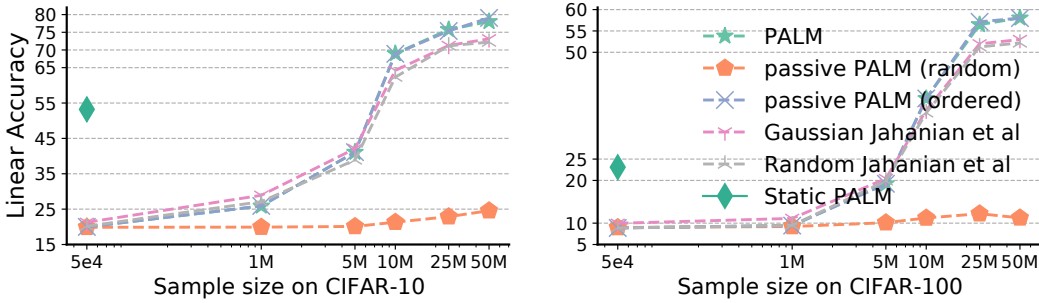

Figure 6: Results of comparing static and passive variants of the proposed method on CIFAR dataset. Sample size denotes the number of images for static PALM and denotes the number of interactions for others. All results are the average of three runs with different seeds.

**Diversity analysis.** In order to measure the diversity of the sampled synthetic data, we randomly sample 10K images from the CIFAR-10 train set and generate 10K images from StyleGAN by random sample latents. We also generate 10K images with PALM in latent space. For each sample in each set, we find its closest neighbor in Inception feature space (obtained after the pooling layer). The coverage is defined as the proportion of nearest neighbors

Table 3: Analysis of diversity of generated data points on CIFAR-100.

| Method | Coverage (%) (higher is better) |
|---|---|
| Latent space uniform sample | 50.16 |
| PALM | 58.87 |

that are unique in the train set. A better active exploration method would produce samples that are equally likely to be close to any image in the train set. PALM achieves a better coverage score than random sampling in latent space. Although the Euclidean distance in Inception feature space is inaccurate, the better coverage achieved by PALM indicates that by exploring the latent manifold it gets more diverse data.

## 7 Conclusion and discussion

Our work presents a method to connect learning from large, diverse static datasets and learning by interaction, and demonstrates promising results of doing so. We believe it opens up an interesting future direction for advancing both reinforcement learning and self-supervised visual representation learning.

For limitations, since it is computational expensive to draw samples from large generative model, interacting with such environment is significantly slower than well optimized physical simulations used in RL, $e.g.$, Mujoco. Addressing such limitation would make this work more accessible and facilitate future research.

For future work, it would be interesting to design or learn better transition dynamics. It would be also interesting to evaluate our method in RL settings where a lot of action-free data are available and won't have to be collected (Minecraft for example). It would be also interesting to use a language conditioned generative model as an alternative $e.g.$ Dalle [57, 58], so the environment has more complexity by prompting it with natural language.

## Acknowledgements

We would like to thank Guillaume Desjardins for insightful discussion and giving constructive comments. We would also like to thank anonymous reviewers for their helpful feedback.

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
