# A  Experiment Details on Representation Pretraining

We follow the same optimization step of SimSiam [11].

The input image size is $32\times32$. We use SGD with base $lr=0.03$ and a cosine decay [46] schedule for 800 epochs, weight decay $=0.0005$, momentum $=0.9$, and batch size $=512$. The encoder $f$ is a ResNet-18 [22] with the last fc layer removed.

We use the same projection and prediction networks in SimSiam. Specifically, the projection MLP $g$ has 3 layers, and has BN applied to each fully-connected (fc) layer, including its output fc. There is no ReLU for the last fc layer in $g$ and the hidden size is 2048. The prediction MLP $(h)$ has 2 layers, and has BN applied to its hidden fc layers. Its output fc does not have BN or ReLU. The dimension of $h$'s input and output is $d=2048$, and $h$'s hidden layer's dimension is 512, making $h$ a bottleneck structure.

After pretraining, the representation encoder $f$ is finetuned for downstream tasks.

# B  Representation Learning Loss and Pseudocode

We use the symmetrized cosine similarity loss from SimSiam. Specifically, for two consecutive observations $s_1$ and $s_2$, denoting the two output vectors as $p_1=h(g(f(s_1)))$ and $z_2=g(f(s_2))$, we minimize their negative cosine similarity:

$$\mathcal{D}(p_1,z_2) = -\frac{p_1}{\|p_1\|_2}\cdot\frac{z_2}{\|z_2\|_2},\tag{1}$$

where $\|\cdot\|_2$ is $\ell_2$-norm. And the loss function is defined as

$$\mathcal{L}=\frac{1}{2}\mathcal{D}(p_1,\texttt{stop\_gradient}(z_2))$$
$$+\frac{1}{2}\mathcal{D}(p_2,\texttt{stop\_gradient}(z_1)).\tag{2}$$

Algorithm 1 shows the pseudocode of representation learning.

# C  Experiment and dataset details

**Datasets**  CIFAR-10 [38] and CIFAR-100 [38] consist of 50,000 training and 10,000 test images with 10 and 20 (super-class) image classes, respectively. For CIFAR-10, out-of-distribution (OOD) samples are as follows: SVHN [49] consists of 26,032 test images with 10 digits, resized LSUN [80] consists of 10,000 test images of 10 different scenes, Interp. consists of 10,000 test images of linear interpolation of CIFAR-10 test images. CelebA [45], a labeled dataset consisting of over $200,000$ face images and each with $40$ attribute annotation. The Atari observation dataset is collected using double DQN [47, 73] with n-step return (n=3) and 3 frames stacking. In total we collected 50M images and each observation is resized to 64x64x3.

**Model details**  For CIFAR10, we use pretrained StyleGAN available at the official website of StyleGAN-Ada[31][2]. This pretrained model is a conditional StyleGAN that achieves best FID score. We also experimented with the model with best Inception score[3] but did not observe significant difference in results.

For CIFAR100, since we did not find publicly available pretrained models, we trained one by ourself using the StyleGAN official code[4].

---

[2]https://nvlabs-fi-cdn.nvidia.com/stylegan2-ada-pytorch/pretrained/paper-fig11b-cifar10/cifar10c-cifar-ada-best-fid.pkl

[3]https://nvlabs-fi-cdn.nvidia.com/stylegan2-ada-pytorch/pretrained/paper-fig11b-cifar10/cifar10c-cifar-ada-best-is.pkl

[4]https://github.com/NVlabs/stylegan2-ada

For Atari, we train an unconditional StyleGAN using the official code. It is difficult to train StyleGAN to converge on our Atari dataset with the default hyperparameters. We found that increasing the batch size to 256 significantly help stabilize the training and accelerate convergence.

**Linear classification**   The quality of the pretrained representations is evaluated by training a supervised linear classifier on frozen representations $h$ in the training set, and then testing it in the validation set. We use batch size of 512, Adam optimizer with learning rate being 0.001. We finetune for 100 epochs in linear classification experiments.

**Atari games**   Our evaluation metric for an agent on a game is *human-normalized score* (HNS), defined as $\frac{agent\_score - random\_score}{human\_score - random\_score}$. We calculate this per game by averaging scores over 100 evaluation trajectories at the end of training, and across 10 random seeds for training. We report both mean (Mn) and median (Mdn) HNS over the 26 Atari-100K games, as well as on how many games a method achieves super-human performance (>H) and greater than random performance (>0).

# D   Experiment details on RL

We adhere closely to the parameter settings from URLB [41], which our implementation is based upon. We list the hyper-parameters in Table 4

We employ clipped double Q-learning for the critic, where each $Q$-function is parametrized as a 3-layer MLP with `ReLU` activations after each layer except of the last. The actor is also a 3-layer MLP with `ReLU`s that outputs mean and covariance for the diagonal Gaussian that represents the policy. The hidden dimension is set to 1024 for both the critic and actor. The actor and critic networks both have separate encoders, although we share the weights of the conv layers between them. Furthermore, only the critic optimizer is allowed to update these weights (e.g. we stop the gradients from the actor before they propagate to the shared conv layers).

Table 4: Hyper-parameters for training the unsupervised RL algorithms.

| Random hyper-parameter | Value |
|---|---|
| Replay buffer capacity | $10^6$ |
| Seed frames | 4000 |
| Mini-batch size | 1024 |
| Discount ($\gamma$) | 0.99 |
| Optimizer | Adam |
| Learning rate | $10^{-4}$ |
| Agent update frequency | 2 |
| Critic target EMA rate | 0.01 |
| Hidden dim. | 1024 |
| Exploration stddev clip | 0.3 |
| Exploration stddev value | 0.2 |
| Reward transformation | $\log(r + 1.0)$ |
| Forward net arch. | $(512 + |\mathcal{A}|) \rightarrow 1024 \rightarrow 512$ ReLU MLP |
| Inverse net arch. | $(2 \times 512) \rightarrow 1024 \rightarrow |\mathcal{A}|$ ReLU MLP |
| $k$ in NN | 12 |
| Avg top $k$ in NN | True |
| Episode length | 200 |

**Algorithm 1** Representation Learning Pseudocode

```
# f: backbone
# g: projection mlp
# h: prediction mlp

for x1, x2 in loader: # load tuples of consecutive observations
    h1, h2 = f(x1), f(x2) # representations, n-by-d
    z1, z2 = f(h1), f(h2) # projections, n-by-d
    p1, p2 = h(z1), h(z2) # predictions, n-by-d

    L = D(p1, z2)/2 + D(p2, z1)/2 # loss

    L.backward() # back-propagate
    update(f, h) # SGD update

def D(p, z): # negative cosine similarity
    z = z.detach() # stop gradient

    p = normalize(p, dim=1) # l2-normalize
    z = normalize(z, dim=1) # l2-normalize
    return -(p*z).sum(dim=1).mean()
```

**Algorithm 2** Environment Pseudocode

```
# P: policy
# G: generator
# alpha, beta: hyperparameters control the importance of action and state.

for t in range(episode_length): #
    z = random_normal() # sample a random latent from Gaussian
    if t == 0: # first step of an episode
        z_t = z
        c = random_label() # sample a random class label
        s_t = G(z_t, c) # generate an observation
        a_t = P(s_t) # sample an action from policy
    else:
        zprime = alpha * a_t + (1 - alpha) * z
        z_t = beta * z_t + (1 - beta) zprime
        s_t = G(z_t, c) # generate an observation
        a_t = P(s_t) # sample an action from policy
```