# OpenReview forum: "Palm up: Playing in the Latent Manifold for Unsupervised Pretraining"
_NeurIPS.cc/2022/Conference — NeurIPS 2022 Accept_

### Official Review · Reviewer_tz3i · 2022-07-09

**Rating:** 7
**Confidence:** 3
**Soundness:** 3 good
**Presentation:** 3 good
**Contribution:** 3 good

**Summary:**

The authors propose to combine RL and GAN to perform unsupervised pretraining. The RL agent's objective is curiosity-based. The authors perform large-scale experiments and analysis to showcase the proposed approach's effectiveness.

**Questions:**

I don't have any specific technical question.

**Limitations:**

The authors adequately discuss the work's limitation.

**Strengths And Weaknesses:**

I want to preface my review by saying that I haven't published any paper in unsupervised pretraining, so I'm not at all an expert in this domain.

Strength:
* I find the idea very interesting, and has a lot of potential. Currently, a simple generator is used. But It's easy to imagine that down the road, more advanced generator can be used to cover the long-tail of data. For example, using text-conditional diffusion models like Dalle / Imagen can drastically help an algorithm to cover the long-tail, which is quite exciting. Though this is not in the scope of the current paper.
* The paper is well written. I can easily follow the paper as a non-expert in unsupervised pretraining.
* The experiment section's analysis seems very solid.

Weakness:
* I think what I'm least convinced about is if the effectiveness comes from the RL part or the GAN part. It's easily to imagine that with the 50M images collected from StyleGAN during RL training, we can use these images as a static dataset and evaluate traditional approaches. With the amount of images generated, it's reasonable to expect that traditional approach might benefit from them. Note that this is different from the "Passive PALM" included in the paper.

---

> ### Author Response · Authors · 2022-08-02
> **Response to reviewer tz3i**
>
> We thank the reviewers for finding our idea very interesting, and has a lot of potentials. At the same time, the reviewer had questions about our passive PALM, which we address below.
>
> > Compare with other traditional ways of using RL collected samples “With the amount of images generated, it's reasonable to expect that traditional approach might benefit from them. Note that this is different from the "Passive PALM" included in the paper.”
>
>
> We thank the reviewer for raising this question. We believe that passive PALM belongs to traditional ways of using RL collected samples, Passive PALM is not trained interactively online. It is trained offline on the experience of PALM.  We first store all the online experience of PALM, then we use the saved experience to train passive PALM without interacting with the environment. There are two variants, the first one is random passive PALM which randomly samples consecutive observations from the stored offline experience. The second one is ordered passive PALM which is trained on the same order of observations as in online experience. Comparing passive PALM with other baselines, indeed we see this approach benefits from more and more RL collected samples.

---

### Official Review · Reviewer_E5Lt · 2022-07-10

**Rating:** 7
**Confidence:** 5
**Soundness:** 3 good
**Presentation:** 2 fair
**Contribution:** 3 good

**Summary:**

This paper casts the latent space of a generative model (StyleGAN) as an _environment_ for unsupervised RL. The agent moves around in latent space seeking out high novelty states, and simultaneously learns a contrastive representation from temporally adjacent observations (generated images). The paper shows that this procedure can serve as effective pretraining for a vision system.

**Questions:**

My questions and suggestions are given above, so here I will just list the two things I would most appreciate in a rebuttal:
1. Material that can convince me that the active sampling component of PALM is more effective than passive ways of sampling training data from StyleGAN.
2. An explanation for the low performance of static PALM.

**Limitations:**

The limitations are adequately discussed. I appreciated the discussion about computational efficiency.

**Strengths And Weaknesses:**

This is such a cool idea! I think the main strength of the paper is that the _idea_ of active representation learning _in generative latent space_ is highly compelling and pretty original. However, the strength of this idea is not convincingly validated by the experiments and results. I will expand on each point below:

**Originality**
The idea is, in my opinion, highly original. It combines several disparate lines of work -- on representation learning, curiosity, and latent variable generative models -- in a new and interesting way. The idea of treating latent space as an environment is really neat. The nearest works I know are those that train latent world models (but these typically do not use off-the-shelf image generators like StyleGAN) and video generative models that learn temporal dynamics in latent space (e.g., https://arxiv.org/abs/2006.10704; but I have not seen these  used for active representation learning). Given the large body of prior work on latent world models for RL environments (e.g., https://arxiv.org/abs/1912.01603), the Atari experiments are arguably not that novel. More interesting to me is the use of a pretrained StyleGAN as an environment. The present work is also quite related to older work in active learning, where the goal was to optimally sample a dataset such that training on those samples minimizes some form of regret. A good summary of that field is given in [Burr Settles. Active learning literature survey. 2009]. I feel this literature should be mentioned in the related work section, but I don't see it as detracting from the novelty of the current paper.

**Quality**
The quality of the paper can be improved in several ways. I will focus on the experiments, as those are the critical failure in my opinion. I think there are three main issues with the experiments:
1. *Section 5 does not yield insight* I didn't take much from Section 5, which compares PALM to other pretraining methods. The comparisons feel like apples to oranges. What do we learn by seeing that PALM (+ data aug) performs about as well as, e.g., SimCLR on CIFAR? These methods differ in innumerable ways: PALM is trained on synthetic data, SimCLR on real data; PALM uses the SimSiam architecture / loss, SimCLR uses a different architecture / loss; etc. The same can be said about pretty much all the comparisons in this section. I have no idea which difference between the methods was critical for explaining why one or the other does better (and anyway, most do about the same). I would suggest moving this entire section to the appendix, or even omitting it from the paper. I understand that the intent might be to show that PALM is competitive with "SOTA" methods, but if that's the goal then this section is still unconvincing. Comparisons on CIFAR and Atari are simply not the setting where one can make compelling claims about SOTA results, because the field, for the most part, is not competing on these benchmarks anymore. I do think these are perfectly fine datasets on which to do _science_ (carefully controlled experiments that yield understanding), but that's not what Section 5 provides.
2. *Missing baselines / ablations* To me, the really interesting experiments are in Section 6. I think this section should be expanded and strengthened. The core novelty of PALM is the active sampling algorithm, so we need to be convinced that it really makes sense and helps. I think the critical experiment would be to compare PALM's sampling method to other ways of sampling from StyleGAN latent space. For example: the samples in Fig 2 look like random draws from the GAN (conditioned on a fixed class). It may be they are not but that suggests a missing baseline: what if you just randomly sample positives from G(z,c=c') (i.e. fix the class, take two random draws from z). Indeed this baseline has been previously published in ref [26] (appendix table 3) and performs quite well there. How does it perform in PALM's experimental setting? I think Fig 4 gets close to testing this -- it's the setting where alpha and beta both equal 0 -- but only a few selected values for alpha and beta are shown. I would like to see all the possible settings (could plot a heatmap of performance with alpha and beta as the axes). [26] also proposes a variety of other ways to sample from GAN latent space, which do not involve active learning but perform well (e.g., positive pairs are given by z and z+w, where w is Gaussian). How does PALM compare to these? The experiments on static PALM are also close to addressing this but they train on real data rather than synthetic data, so again, it's apples to oranges and unclear if the active learning is what's helping. Passive PALM was hard for me to interpret since the data is still initially collected in an active manner.
3. *Questionable baselines* The performance of static PALM is suspicious to me. If I understand correctly, passive PALM is essentially SupCon but with the SimSiam architecture / loss. Why does it perform so poorly on CIFAR? The SupCon paper reports 96% accuracy on CIFAR10, whereas static PALM gets <55%. That gap is too large for me to believe that static PALM is a compelling baseline. Yes, the SupCon method differs in many ways but it nonetheless demonstrates that static sampling on real data can be very effective. I would like to better understand why the PALM numbers are so low in Fig 5. My concern is that the methods have not been tuned well or trained sufficiently to make meaningful conclusions.


**Clarity**
The paper is mostly clear but 1) the latent dynamics model came across as ad hoc; can it be better motivated? 2) What is a tanh Gaussian (line 206)?

**Significance**
I think the significance is high if the method really works. I'm not yet convinced it does, but I do believe in the framework. As more powerful generative models become available (dall-e, etc), I think the direction explored here might turn out to be increasingly important.

---

> ### Author Response · Authors · 2022-08-02
> **Response to reviewer E5Lt**
>
> We thank the reviewer for finding our idea cool, highly compelling and pretty original. At the same time, the reviewer had concerns about missing baselines and results of an ablated baseline, which we address below.
>
> > Material that can convince me that the active sampling component of PALM is more effective than passive ways of sampling training data from StyleGAN.
>
>
> We thank the reviewer for suggesting two baselines. The first one is randomly sampling positives from G(z,c=c') (i.e. fix the class, take two random draws from z). We denote this method as Random Jahanian et al. The second one is positive pairs are given by z and z+w, where w is Gaussian. We denote this method as Gaussian Jahanian et al.
> The results are shown in Figure 6, Section 6. With 50M synthesized CIFAR-10 samples and using the same setting as PALM, the random method achieves 72.23 and the Gaussian method achieves 73.15, for reference, PALM achieves 78.04 accuracy. The results show that while methods from [26](Jahanian et al) perform surprisingly well compared to other baselines, PALM performs substantially better.
> In the low data regime, carefully designed non-active learning based methods from [26](Jahanian et al) outperforms PALM, possibly due to RL training being sample inefficient. With the increasing number of synthesized samples, PALM continues to improve and significantly outperform all baselines. The results also show that PALM scales better than non active learning based methods.
>
> > I would like to see all the possible settings (could plot a heatmap of performance with alpha and beta as the axes)
>
>
> We thank the reviewer for suggesting this experiment. We present the results of more combinations of $\alpha$ and $\beta$ in Figure 5, Section 5. The results show that as $\alpha$ decreases from 0.95 to 0, because of the decreasing importance of action, the accuracy continues to drop. $\alpha=0.95$ in general performs better than $\alpha=1.0$, indicates mixing a small amount of random noise with action helps exploration and improves representation learning. When $\alpha$ or $\beta$ is zero, the results are significantly worse. We also see that $\beta \in [0.5, 0.75]$ performs significantly better than other values, showing that persistence of dynamics is crucial.
>
> > An explanation for the low performance of static PALM.
>
>
> While static PALM is related to SupCon, as the reviewer mentioned, it differs in many ways. Let us present a study to explain them. Firstly, static PALM does not use any data augmentation while SupCon leverages a combination of random resize, random crop, grayscale, and color jitter, if we combine SupCon data augmentation with static PALM, it gets 7.5 percent improvement. Secondly, SupCon uses a large number of negative examples from different categories, which significantly help contrastive learning. Combine this with static PALM, it boosts the accuracy by 13.6. Finally, there are also differences in projection head, learning rate, and number of training epochs.
>
> We hope that our reply addresses all of your questions, but if not, please let us know.

---

> > ### Comment · Reviewer_E5Lt · 2022-08-08
> > **thanks for the new experiments; raising score**
> >
> > Thanks for running the experiments I asked for! I'm much more convinced now that the method is working. Random sampling is a hard baseline and I believe many active learning methods don't, in practice, outperform random. So I find the new results on CIFAR at 50M, outperforming the Jahanian et al. baselines, to actually be quite strong.
> >
> > The new diversity analysis is also interesting and compelling.
> >
> > I still find static PALM to be a bit weak as a baseline. It would be interesting to run all the methods in Section 6 with data aug -- i.e. PALM+data aug, static PALM+ data aug, etc -- to see if PALM still holds up in this more practical setting.
> >
> > On the writing, I would still recommend reducing Section 5 and instead expanding Section 6 even more, but that's somewhat subjective. I find Section 6 much more interesting.
> >
> > Since my critical questions were addressed, and since I really like the idea, I now think this paper should be accepted!

---

> > > ### Author Response · Authors · 2022-08-08
> > > **Thank you**
> > >
> > > We thank the reviewer for replying and the feedback that helped us make this paper better.
> > >
> > > We will address the suggestions in the final version of this paper.

---

### Official Review · Reviewer_ja7N · 2022-07-11

**Rating:** 3
**Confidence:** 4
**Soundness:** 1 poor
**Presentation:** 1 poor
**Contribution:** 1 poor

**Summary:**

This paper proposes a method that exploits both the large diverse datasets and exploratory behavior. Authors used the generator that was pre-trained on the static datasets, while introducing the dynamic model in the latent space. Transition dynamics are modeled via mixing an action and a latent variable. Further, temporal persistence is exploited via moving average. Experiments on several down-stream tasks demonstrate the effectiveness of the learned representation.


**Questions:**

In Fig. 3, I suggest using a finer-grained y axis grid.
I suggest authors conduct experiments in more useful and practical situations.

**Ethics Review Area:**

["I don’t know"]

**Strengths And Weaknesses:**

(+) Authors proposed interesting high-level concepts that try to mix dynamic exploration and the latent variables. The idea itself looks intuitive and intuitive.
(-) The actual implementation of mixing dynamic exploration and the latent variable is quite heuristic. It is the simple weight summation of the two variables. Also, the temporal persistence module is similar. The overall accuracy also varies a lot via the heuristic parameters alpha and beta as shown in Fig. 4.
(-) The experiments are conducted on several down-stream tasks; while I cannot see what are the benefits when performing the experiments on such domains. I encourage authors to conduct the experiments in more useful and practical situations.
(-) I cannot understand what trajectories and time axes mean in image classification tasks. It seems that no temporal information exists in such an application; while authors still use the temporal persistency module. Furthermore the accuracy gain is not appealing.

---

> ### Author Response · Authors · 2022-08-02
> **Response to reviewer ja7N**
>
> We thank the reviewer for finding our ideas interesting and intuitive. At the same time, the reviewer had concerns about the motivation for using a linear dynamics model and about our experimental setup, which we address below.
>
>
> > The actual implementation of mixing dynamic exploration and the latent variable is quite heuristic. It is the simple weight summation of the two variables. Also, the temporal persistence module is similar. The overall accuracy also varies a lot via the heuristic parameters alpha and beta as shown in Fig. 4.
>
>
>
> While our linear dynamics module is somewhat limited, it is quite common in robotics [1, 2]. Persistent representations are also quite popular in learning representations from videos by minimizing different metrics of representation difference over a temporal segment[3, 4], and  in reinforcement learning it has been demonstrated to improve data efficiency and improve downstream task performance [5, 6]. Lastly, we would also like to point out that Fig. 4 demonstrates the importance of having the persistence module.
> [1] Watter, Manuel, et al. "Embed to control: A locally linear latent dynamics model for control from raw images." Advances in neural information processing systems 28 (2015).
> [2] Zhang, Wenbo, et al. "Deformable linear object prediction using locally linear latent dynamics." 2021 IEEE International Conference on Robotics and Automation (ICRA). IEEE, 2021.
> [3] S. Becker. Learning temporally persistent hierarchical representations. In Advances in neural information processing systems, pages 824–830. Citeseer, 1997.
> [4] L. Wiskott and T. J. Sejnowski. Slow feature analysis: Unsupervised learning of invariances. Neural computation, 14(4):715–770, 2002.
> [5] M. Schwarzer, A. Anand, R. Goel, R. D. Hjelm, A. Courville, and P. Bachman. Data-efficient reinforcement learning with self-predictive representations. In International Conference on Learning Representations, 2021.
> [6] P. Sermanet, C. Lynch, Y. Chebotar, J. Hsu, E. Jang, S. Schaal, S. Levine, and G. Brain. Time-contrastive networks: Self-supervised learning from video. In 2018 IEEE International Conference on Robotics and Automation (ICRA), pages 1134–1141. IEEE, 2018.
>
> > The experiments are conducted on several down-stream tasks; while I cannot see what are the benefits when performing the experiments on such domains. I encourage authors to conduct the experiments in more useful and practical situations
>
>
> Pretraining has been the cornerstone of many impressive successes at the frontier of artificial intelligence, such as understanding natural language, reinforcement learning, and image recognition [1, 2, 3, 4, 5, 6]. We believe that considering image classification and reinforcement learning as downstream tasks is appropriate for proof of concept, although we do agree that more useful and practical domains would be interesting.
>
> [1] Brown, Tom, et al. "Language models are few-shot learners." Advances in neural information processing systems 33 (2020): 1877-1901.
> [2] Reed, Scott, et al. "A generalist agent." arXiv preprint arXiv:2205.06175 (2022).
> [3] Dosovitskiy, Alexey, et al. "An image is worth 16x16 words: Transformers for image recognition at scale." arXiv preprint arXiv:2010.11929 (2020).
> [4] Baker, Bowen, et al. "Video PreTraining (VPT): Learning to Act by Watching Unlabeled Online Videos." arXiv preprint arXiv:2206.11795 (2022).
> [5] Radford, Alec, et al. "Learning transferable visual models from natural language supervision." International Conference on Machine Learning. PMLR, 2021.
> [6] Devlin, Jacob, et al. "Bert: Pre-training of deep bidirectional transformers for language understanding." arXiv preprint arXiv:1810.04805 (2018).
>
> We hope that our response addresses all of your questions, but if not, please let us know.

---

> > ### Comment · Reviewer_ja7N · 2022-08-09
> > **Response to author rebuttal**
> >
> > Authors have dealt with most of my concerns effectively. I have nore more questions at this stage.

---

### Official Review · Reviewer_bFfS · 2022-07-11

**Rating:** 6
**Confidence:** 3
**Soundness:** 3 good
**Presentation:** 3 good
**Contribution:** 3 good

**Summary:**

This paper introduces an RL-based exploration strategy to the generative models that have been trained on very large datasets. The proposed approach aims to build a world model by learning a transition function in the latent space of the generative models. It is then used to train an agent interactively in this "environment", which aims at exploration along with unsupervised representation learning. The paper evaluates the learned representations in a number of downstream tasks such as classification, out-of-distribution detection, and representation learning in RL, and presents promising results.

**Questions:**

1. For a given representation $h_i$ in the exploration phase, the representations in the temporal vicinity of $h_i$ (i.e., $h_{i-1}$, $h_{i+1}$, etc.) should be the ones in the nearest neighbor set according to the Siamese network’s objective. However, the exploration rewards the discrepancy between them. How do the Siamese network objective and the unsupervised exploration objective work coherently?

2. The proposed interaction mechanism is not fully clear to me. What is the purpose of using a random latent sample? Despite the exponential moving average, the updated latent $z_{t+1}$ may produce a significantly different observation $s_{t+1}$ than $s_t$, particularly in the unconditional setting. The predicted action is conditioned on only the state $s_t$ and the policy is unaware of the random latent sample. Could the authors clarify why the proposed approach is used? Couldn’t the policy be conditioned on the random sample?


**Limitations:**

Yes.

**Strengths And Weaknesses:**

Strengths
1. The paper is well-motivated and well-written.
2. The idea of leveraging a large generative model as an environment for diverse data sampling is very interesting.
3. Evaluations are performed on multiple tasks in various domains.
4. Although it is proposed for unsupervised and task-agnostic representation learning, it could also be augmented with task-based rewards to develop a task-related sampling procedure.

Weaknesses
1. This feels like a chicken and egg problem in some cases. As it is presented in the Atari experiment, the generative model requires a large number of samples (50M) in the first place. The proposed model requires only 6M samples from it. It seems highly inefficient. What would be the performance if the Atari policy was pre-trained on all the 50M samples and also on a subset with 6M samples, and then fine-tuned?
2. The paper does not present an evaluation of the diversity of the sampled synthetic data points.
3. I think another problem is with the following statement in line #112: "Different from them, our work focuses on leveraging a generative model as an interactive environment and eliminating the need for data augmentation." In the results in Fig. 3, it is clear that the proposed model is not a replacement for augmentation. Maybe the contribution claims regarding the augmentation should be toned down.

---

> ### Author Response · Authors · 2022-08-02
> **Response to reviewer bFfs**
>
> We thank the reviewer for finding our ideas well-motivated and very interesting. At the same time, the reviewer had questions about missing evaluation of data diversity and about atari data efficiency, which we address below.
>
>
> > The paper does not present an evaluation of the diversity of the sampled synthetic data points.
>
> We thank the reviewer for suggesting measuring the diversity of the sampled synthetic data. We now report nearest neighbor based coverage in Table 6. We randomly sample 10K images from the CIFAR-10 train set and generate 10K images from StyleGAN by random sample latents. We also generate 10K images with PALM in latent space. For each sample in each set, we find its closest neighbor in Inception feature space (obtained after the pooling layer). The coverage is defined as the proportion of nearest neighbors that are unique in the train set. A better active exploration method would produce samples that are equally likely to be close to any image in the train set. PALM achieves a better coverage score than random sampling in latent space. Although the Euclidean distance in Inception feature space is inaccurate, the better coverage achieved by PALM indicates that by exploring the latent manifold it gets more diverse data.
> > In Atari experiments, training the generative model takes a large number of samples which is inefficient.
>
> We thank the reviewer for asking this question. We believe that in the near future there are going to be applications in which a lot of data will be available and won't have to be collected (Minecraft for example). Thus, the data collection process in the Atari experiments should be thought of as a proof of concept. We picked Atari for this simply because it is a common benchmark. A more reasonable application of our method would utilize existing data/generative models and won’t have to interact with the environment to collect data.
>
> > How do the Siamese network objective and the unsupervised exploration objective work coherently?
>
> The unsupervised exploration objective encourages exploring in the abstract representation space. The representation is learned by the Siamese network objective and unsupervised exploration objective does not update the representation. This is pretty common in unsupervised and self-supervised RL [1, 2, 3].
>
> [1] Srinivas, A., Laskin, M. and Abbeel, P., 2020. Curl: Contrastive unsupervised representations for reinforcement learning. arXiv preprint arXiv:2004.04136.
> [2] Schwarzer, Max, et al. "Pretraining representations for data-efficient reinforcement learning." Advances in Neural Information Processing Systems 34 (2021): 12686-12699.
> [3] Liu H, Abbeel P. Aps: Active pretraining with successor features. InInternational Conference on Machine Learning 2021 Jul 1 (pp. 6736-6747). PMLR.
>
> > What is the purpose of using a random latent sample? The policy is unaware of the random latent sample. Could the authors clarify why the proposed approach is used? Couldn’t the policy be conditioned on the random sample?
>
> Adding random latent noise to action is empirically chosen, in our experiments, we found that introducing environment noise improves downstream performance although it could make unsupervised exploration more difficult. In our special case of linear dynamics, this is similar to adding exploration bonus like epsilon greedy. Another interpretation is that it is mixing the random latent model with the linear dynamics.

---

### Author Response · Authors · 2022-08-02
**General response**

We thank all reviewers for their constructive comments and suggestions. We are glad to see that all reviewers generally appreciate our results, novelty and originality, and think the idea “well-motivated and very interesting” (bFfs), “interesting and intuitive” (ja7N), “cool, highly compelling and pretty original” (E5Lt), “very interesting, and has a lot of potentials” (tz3i).

We address the concerns each reviewer has in the individual responses below.

---

### Meta-Review · Area_Chair_FoJs · 2022-08-30

**Recommendation:** Accept
**Confidence:** Certain

**Metareview:**

This paper formulates a framework that uses the latent space of pretrained generative models, particularly StyleGAN, as the environment for a dynamics model (unsupervised RL). The RL agent explores the latent space decoding actions and random latents to images through the pretrained generator in an unsupervised fashion, learning representations evaluate on downstream tasks.

The reviewers agreed that the paper has clear motivation and is well written, and that the idea has clear novelty and originality.
The only issues raised are to do with a few more informative baselines and comparisons, and on evaluating the diversity of samples in the trajectories, which the authors address with additional experiments and discussion in the updated manuscript.

The paper clearly has merit and explores an original idea with a good set of experiments (post rebuttal) that cover a range of settings, and should be accepted.
Having said that, I would strongly urge the authors to take comments on editing Sec5 into consideration, and also ensure that the manuscript is still within the intended page limits with the edits.


**Award:**

No

---

### Decision · Program_Chairs · 2022-09-14

Accept